# DutaFabs are engineered therapeutic Fab fragments that can bind two targets simultaneously

Roland Beckmann 1✉, Kristian Jensen[1,4], Sebastian Fenn[1], Janina Speck[1], Katrin Krause[1], Anastasia Meier[1], Melanie Röth[1], Sascha Fauser[2], Raymond Kimbung[3], Derek T. Logan[3], Martin Steegmaier[1] & Hubert Kettenberger[1]

We report the development of a platform of dual targeting Fab (DutaFab) molecules, which comprise two spatially separated and independent binding sites within the human antibody CDR loops: the so-called H-side paratope encompassing HCDR1, HCDR3 and LCDR2, and the L-side paratope encompassing LCDR1, LCDR3 and HCDR2. Both paratopes can be independently selected and combined into the desired bispecific DutaFabs in a modular manner. X-ray crystal structures illustrate that DutaFabs are able to bind two target molecules simultaneously at the same Fv region comprising a VH-VL heterodimer. In the present study, this platform is applied to generate DutaFabs specific for VEGFA and PDGF-BB, which show high affinities, physico-chemical stability and solubility, as well as superior efficacy over anti-VEGF monotherapy in vivo. These molecules exemplify the usefulness of DutaFabs as a distinct class of antibody therapeutics, which is currently being evaluated in patients.

[1] Roche Pharma Research and Early Development, Roche Innovation Center Munich, Roche Diagnostics GmbH, Penzberg, Germany. [2] Roche Pharma Research and Early Development, Roche Innovation Center Basel, F. Hoffmann-La Roche Ltd, Basel, Switzerland. [3] SARomics Biostructures AB, Medicon Village, Lund, Sweden. [4] Present address: Bornspire, Copenhagen, Denmark. ✉email: pred.dutafabs@roche.com

As monoclonal antibodies continue to play a central role in the development of new large molecule therapeutics, an increasing number of bispecific antibody approaches has been developed and described in the literature[1]. Bispecific monoclonal antibodies can perform a wide range of complex mechanisms of action across numerous therapeutic indications, thereby going beyond the mechanisms possible with monospecific protein drugs[2]. Engineered human antibodies typically carry one specificity per heterodimeric Fv region comprising one VH and one VL variable domain, and bispecific antibody formats typically combine two different Fv regions as building blocks into one molecule[1]. IgG-like bispecific constructs have been described that do not comprise regular Fab arms, but engineered arms where two VHH domains are fused to the CH1 and CK constant domains in place of the regular heterodimeric Fv[3]. Finally, a few examples of bispecific Fv regions comprising a VH-VL hetero-dimer have been reported: Novimmune described engineered Fv regions, each specific for two non-homologous chemokines that are bispecific based on structural mimicry of two targets[4,5], Genentech described dual action Fabs that carry Fv regions with partially overlapping paratopes against two unrelated targets[6,7], and GSK/Domantis, as well as Abbott/AbbVie published Fv regions where the VH and the Vk domains are essentially two distinct single-domain antibodies directed against two different cytokines[8–10].

However, to date no bispecific system is available where two distinct and spatially separated paratopes are presented within the CDRs of the same human, heterodimeric Fv region in such a manner that they can co-bind their targets simultaneously, and are structurally independent from one another. The simultaneous binding of two targets to such a molecule would allow special therapeutic mechanisms, including neutralizing two target molecules with one regular Fab fragment or bringing two immune receptors into very close proximity. Furthermore, combining two

different bispecific Fabs with such characteristics into one molecule would result in antibodies structurally equivalent to conventional IgGs that are trispecific or tetraspecific. Independent optimization of each paratope would enable the routine development of such therapeutic compounds.

This work outlines the development of a platform for generating dual targeting antibody Fab fragments (DutaFabs), where two separate paratopes specific for two distinct targets or epitopes are formed by the CDRs of one human Fv region. We discovered a configuration where the two paratopes are structurally independent, allowing them to each be matured to high affinities without impact on the other paratope. The DutaFab geometry provides spatial separation of the two paratopes sufficient to allow for simultaneous binding of two targets to the same Fab fragment when desired. We conclude that DutaFabs exhibit excellent drug-like properties, have delivered in vivo proof-of-concept, and are ideally suited to clinical development, particularly in the field of ophthalmology.

## Results

**Geometric considerations.** Upon inspection of many Fab structures, we noticed that two distinct areas within the human CDR loops can be defined which are located on opposite sides of the Fv and are potentially suitable for independent target binding: first, a region located towards the VH N-terminus and comprising HCDR1, HCDR3, and LCDR2 residues (hereinafter referred to as H-side paratope; red in Fig. 1a), and second, a region located towards the VL N-terminus and comprising LCDR1, LCDR3, and HCDR2 residues (hereinafter referred to as L-side paratope; blue in Fig. 1a). This observation raises the possibility to separately select a monospecific Fab fragment binding a first target via the H-side paratope and another monospecific Fab fragment binding a second target via the L-side

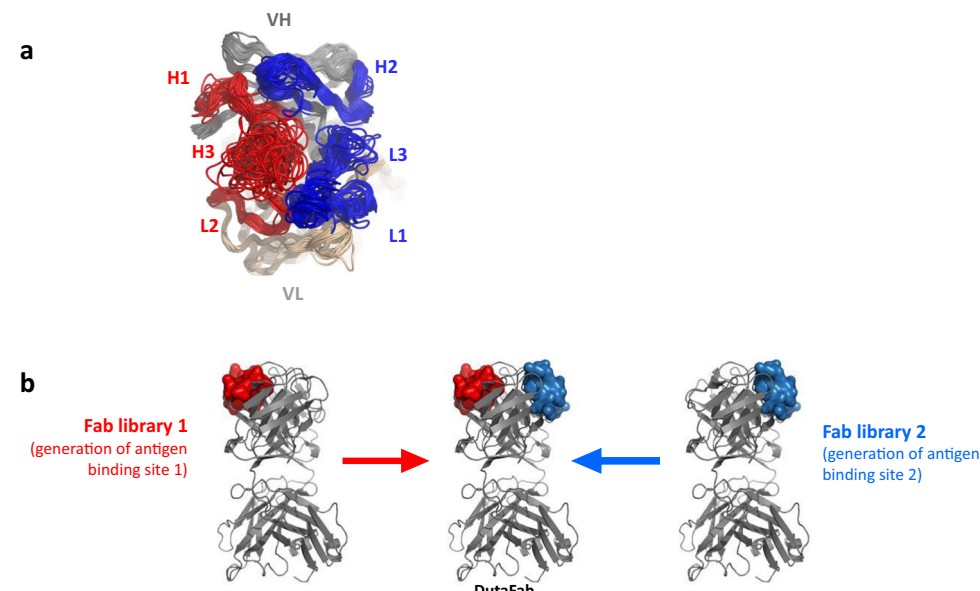

**Fig. 1 The concept of DutaFab paratopes. a** The DutaFab concept of separating paratopes on a single Fab. The structure shows a top-down view of 133 overlaid Fab homology models of late-stage clinical antibodies from Jain et al.[11] It is visible that CDRs 1 and 3 from one chain plus CDR 2 from the other chain each form a separate, contiguous surface. The H-side paratope shown in red comprises HCDR1, HCDR3, and LCDR2, while the L-side paratope shown in blue comprises LCDR1, LCDR3 and HCDR2. The structurally diverse HCDR3 is restricted in length and conformation in DutaFab library designs to ensure modularity. **b** Discovery principle of DutaFabs. Individual paratopes are isolated by separate panning of monospecific Fab phage display libraries, wherein either the H-side or the L-side is diversified in a subset of residues of the relevant CDRs. DutaFabs are generated by gene synthesis and expression of Fab fragments that comprise all the target-specific paratope residues against both targets.

paratope, and then combine the two monospecific binders into a single DutaFab that carries both paratopes and is bispecific for the two targets (Fig. 1b).

Figure 1a shows an overlay of 133 homology models, representing the therapeutic antibodies in the INN collection[11]. These homology models illustrate that the typical shape of L-side paratopes in most human Fabs is flat or somewhat convex with little variation in overall geometry; it differs greatly only if very long LCDR1 sequences are used, as found in the human Vk2 and Vk4 families. The shape of H-side paratopes depends very much on the type and length of HCDR3; the homology models illustrate the large diversity of shapes resulting from different HCDR3 lengths and sequences. To allow the modular construction of bispecific Fab fragments from two paratopes, a more homogenous collection of Fab structures seemed desirable. To achieve this, we utilized short HCDR3s of the kinked conformation type, comprising an R/K94 and D101 residue pair (Kabat numbering)[12–14]. Such HCDR3s result in a somewhat pocket-shaped H-side paratope. For a bispecific platform, having the combination of both flat/convex and pocket-shaped paratopes available may allow for better epitope coverage and hence it was decided to combine a human Vk germline family with a short LCDR1 on the L-side with a design resulting in a short and kinked HCDR3 on the H-side.

**Development of a highly stable human VH3-Vk1 scaffold.** For a robust platform, it is crucial to develop a system where the L-side and H-side paratopes can be combined in a modular and independent manner. It was hypothesized that a highly thermostable VH-Vk pairing could facilitate the desired independent behavior of H-side and L-side paratopes and thus facilitate modularity of the DutaFab platform.

For high thermostability of human Fab fragments two aspects are potentially critical. First, the individual folding stability of both VH and Vk domain needs to be high. Consequently, the DutaFab platform is based on pairings of human VH3 and Vk1 germline genes, as these families are abundant in the expressed repertoire in humans[15] and utilized by several approved antibody drugs[11]. The family-specific residues responsible for the stable folding of the VH3 and Vk1 domains are well known[16,17]. Second, the interface residues between VH3 and Vk1 domains impact their stable packing. Biophysical properties of Fab fragments and their thermal stabilities vary widely[18], and especially in the presence of conserved framework regions, such wide variation can logically be attributed to CDR residues influencing interface packing and folding stability. However, to date no comprehensive description of the influence of CDR residues on human Fab stability exists; therefore, we decided to map the influence of different amino acids and diversification in numerous potentially important CDR positions on the overall stability of human VH3-Vk1 Fab fragments. This CDR mapping allows for educated design of monospecific H-side and L-side libraries to enable the selection of paratopes that can later be combined in a modular manner. Numerous CDR positions were investigated by making individual point mutants in germline-like VH3-Vk1 Fabs and determining the influence on thermostability using differential scanning calorimetry (DSC). In total, hundreds of mutations were investigated with the results being summarized in a patent application (WO2012163519). The most significant influence on Fab stability could be attributed to the interface between HCDR3 and LCDR3, especially positions VH97 and Vk91, to the interface between LCDR1 and LCDR2, in particular position Vk34, and to the portion of HCDR2 nearest to HCDR1, including position VH50.

The findings of this CDR mapping were incorporated into the design of VH3-Vk1 DutaFab library dummies that have sequences close to human germline but include the above stabilizing CDR features. Interestingly, combining the stabilizing features resulted in very thermostable DutaFabs. The highest thermostabilities were observed when interface CDR residues VH35, VH50, VK34, and VK91 were conserved as the invariant residues shown in Fig. 2c. We found that the stabilized dummy Fabs melted in a co-operative manner up to stabilities with a Fab melting temperature ($T_m$) of around 99 °C, but in case of Fvs with an even higher $T_m$ (up to 106 °C was observed), a distinct earlier unfolding event occurs at approximately 95 °C. We conclude that this event represents the independent unfolding of the CH1-Ck constant domain pair within the Fab fragment in the presence of folded Fv (Fig. 2a). i

**Diversification and library construction.** In order to generate DutaFabs, matching pairs of phage display libraries were diversified, wherein each pair was based on the same VH3-Vk1 scaffold (Fig. 2c). Several design steps were taken to retain high thermostability throughout the library members. The most significant stabilizing CDR residues identified in the CDR mapping work were kept invariant or restricted to include only the most stable amino acids for that position. HCDR3 was always diversified but restricted to one length per library to eliminate the possibility of length variation having a detrimental impact on the L-side paratope. Moreover, all HCDR3s were designed in the kinked conformation with an invariant D101 and a basic residue in position VH94. Finally, position 99 in the HCDR3, as well as position 49 directly in front of LCDR2 were each restricted to Tyr and Phe, to allow formation of an aromatic pair between these positions that is frequently found in natural antibodies and stabilizes HCDR3 conformation. Figure 2b shows the location of these stabilizing and conformational CDR residues in the Fv structure.

As illustrated in Fig. 1b and further detailed in Fig. 2c, each diversified pair comprises two distinct libraries for the isolation of monospecific paratopes, intended to be combined into bispecific DutaFabs. One library in the pair is diversified in the region potentially suitable as H-side paratope (HCDR1, HCDR3, and LCDR2), while the other library is diversified in the region potentially suitable as L-side paratope (LCDR1, LCDR3, and HCDR2). In some naïve libraries we also diversified a small number of residues in the frameworks potentially suited to enhance affinity or specificity, primarily by influencing the conformation of the CDRs. These enhancing residues are located in the VH outer loop (which interacts with HCDR1) and at the VH N-terminus in the case of H-side libraries (light red markings), and located in the Vk outer loop (which interacts with LCDR1) and at the Vk N-terminus in the case of L-side libraries (light blue markings). The phage display libraries routinely reach sizes of >$10^{11}$ independent transformants and the combined diversity of DutaFab libraries available now is >$10^{12}$, which compares favorably to other antibody libraries used in the industry[19,20].

**Panning, screening, and success rate.** Naïve monospecific DutaFab libraries diversified in the H-side or L-side paratope were separately selected against various protein targets, following standard phage display protocols. Panning was typically performed for 4 rounds, and the polyclonal selection output of rounds 3 and 4 was converted into vectors for soluble Fab expression. Individual clones of monospecific DutaFabs were expressed for high throughput screening. Such selections were

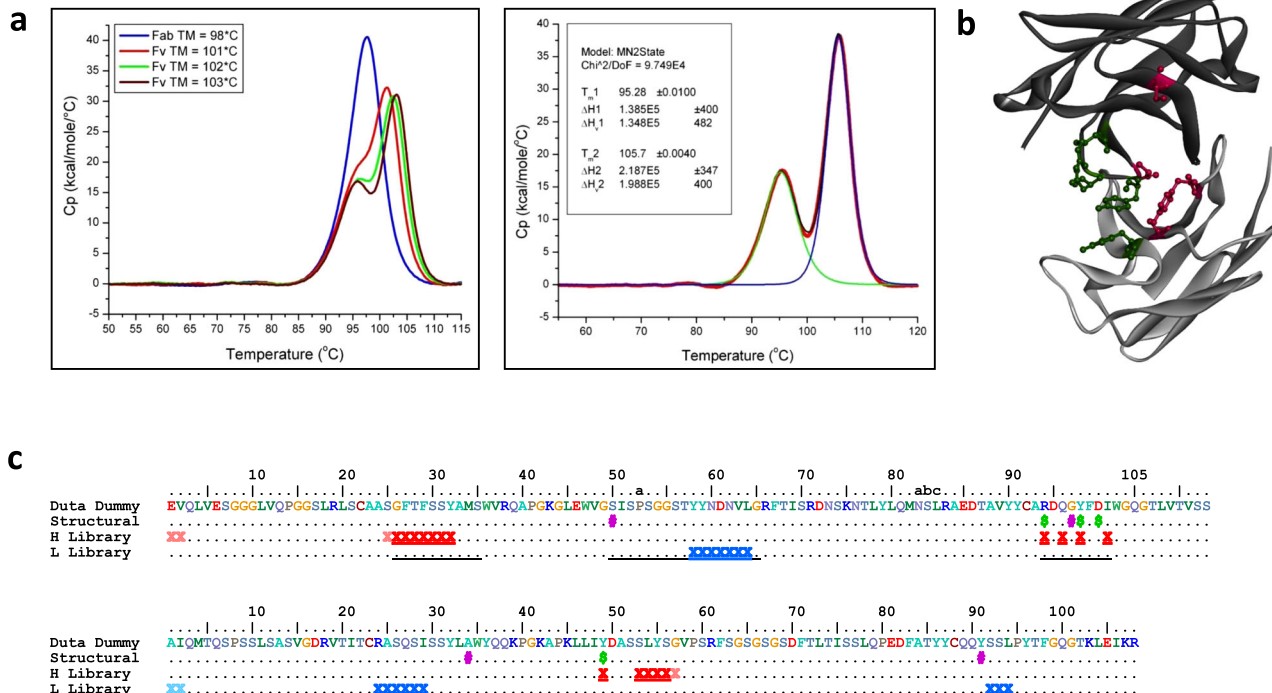

**Fig. 2 DutaFab libraries. a** DSC data of different human VH3-Vk1 DutaFab dummy scaffolds. Left side: Up to a Fab thermostability of 99 °C, co-operative melting of all 4 domains in the Fab is observed (blue trace). Above 99 °C, a left shoulder starts to appear, representing the unfolding of CH1-Ck constant domains in the presence of folded Fv (red, green and brown traces of Fab unfolding data). Right side: In case of high Fv stabilities, constant domain unfolding can be analyzed independently; the red trace represents the data for a Fab fragment with an Fv TM of 105.7 °C, the green trace the non-2-state fit for the earlier melting constant domain unfolding event and the blue trace the non-2-state fit for the later melting Fv unfolding event (fitted with Origin software). **b** Top-down view on a DutaFab scaffold showing conserved structural residues that maintain stability or conformation. DutaFab heavy and light chains are colored dark and light gray, respectively. Residues conferring high stability or restricting HCDR3 conformation are colored according to panel **c**. The structure is based on PDB 6T9D, which is discussed in further detail below. **c** Typical DutaFab library design, showing H-side diversified residues (dark red X), optionally diversified H-side enhancing residues (light red X), L-side diversified residues (dark blue X), optionally diversified L-side enhancing residues (light blue X), stability residues (pink #) and HCDR3 conformational residues (green $). Underlined: all positions designated as CDRs in either the Kabat, Chothia, and Contact definitions[43-45].

performed using the monospecific H-side and L-side libraries on more than 20 targets, including various soluble proteins and eight membrane proteins. Specific binders could be raised for all targets and between 20 and several hundred unique specific binders were found for most targets. The number of unique specific binders was usually similar for H-side and L-side libraries. In our hands, this success rate is similar to that found for conventional monospecific antibody libraries where all 6 CDRs can contribute to a single paratope, and it clearly demonstrates that both combinations of only 3 CDRs investigated here offer paratope surfaces suited to specifically binding a wide range of protein targets and epitopes.

**Generation of bispecific DutaFabs**. To generate bispecific DutaFabs, one pair of an H-side monospecific binder against a first target and an L-side monospecific binder against a second target was chosen per bispecific candidate. The bispecific protein sequence was then generated in silico by making an alignment of the germline-like scaffold sequence with the H-side binder and the L-side binder, and substituting all potential paratope residues from both binders into the scaffold sequence. Potential paratope positions were considered to be all residues in HCDR1, HCDR3, LCDR2, the VH outer loop and at the VH N-terminus for H-side paratopes, and all residues in LCDR1, LCDR3, HCDR2, the Vk outer loop and at the Vk N-terminus for L-side paratopes (see Fig. 2c). Crucially, when following this simple paratope merging procedure, we found that specific target binding for each paratope

was nearly always retained after conversion of the monospecific Fab into a DutaFab, and the affinity and kinetic binding characteristics for each target were generally retained. To date, bispecific DutaFabs against 20 different pairings of protein targets were generated by combining L-side paratopes directed against 14 different protein targets with H-side paratopes directed against 18 different protein targets, using some paratopes in more than one bispecific molecule. Remarkably, of these 20 different protein target combinations, in 19 antibodies the affinities of the H-side and L-side binders were essentially retained with binding affinities against both targets typically within twofold and always within tenfold of the parental monospecific affinities, while one protein target combination required optimization. In projects where highest affinities were desired, the merging procedure was typically followed by further optimization, which is also often required for conventional monospecific antibody projects. This optimization was typically performed in the bispecific DutaFab context, by re-diversification of one paratope in the presence of an invariant opposite paratope.

**Targeting VEGFA and PDGF-BB in macular diseases**. In age-related macular degeneration and diabetic macular oedema, anti-VEGF therapy has been a breakthrough success since the approval of ranibizumab[21] and this area continues to grow[22,23]. However, it would be desirable to find treatments that provide better long-term efficacy than anti-VEGF therapy alone, potentially through bispecific antibody drugs[24]. One therapeutic target

that may offer additional efficacy in the context of anti-VEGF therapy is PDGF-BB. PDGF-BB plays an important role in the proliferation and survival of pericytes surrounding the endothelial cells of sprouting blood vessels[25]. Anti-PDGF-BB therapy may inhibit pericyte development and destabilize sprouting vessels, thus reducing choroidal neovascularization. Furthermore, pericytes deliver VEGF to endothelial cells in a close paracrine manner, resulting in a fraction of VEGF that is inaccessible to conventional anti-VEGF monotherapies, regardless of their retinal penetration, thus creating a potential resistance mechanism to monospecific anti-VEGF drugs[26–28]. Therefore, anti-PDGF-BB therapy may offer additional efficacy beyond anti-VEGF monotherapy based on at least two mechanisms.

Clinically, the hypothesis of VEGFA and PDGF-BB combination therapies received attention following positive phase II trial data suggesting that co-injection of ranibizumab (anti-VEGFA) and pegpleranib (Fovista®, anti-PDGF-BB) was superior to ranibizumab monotherapy[29]. Since then, however, this particular combination therapy has failed to show statistically significant benefits over anti-VEGF monotherapies in phase III trials[30]. Pegpleranib is a PEGylated aptamer, and because its retinal penetration properties have not been characterized or described in detail, it is unclear whether pegpleranib actually efficiently reached the site of neovascularization in patients. Therefore, therapy with a highly potent bispecific Fab fragment having the same molecular architecture as ranibizumab and

comparable retinal penetration properties would seem an attractive option.

**DutaFab against VEGFA and PDGF-BB.** To generate a DutaFab against VEGFA and PDGF-BB, monospecifc phage library selections were performed independently on the two targets. An L-side binder for VEGFA and an H-side binder for PDGF-BB were paired to result in an initial bispecific DutaFab (clone VP), with a variable region amino acid sequence as shown in Supplementary Fig. S1. As illustrated in Fig. 3, clone VP already had moderate to high potencies with an IC$_{50}$ of 21 nM towards PDGF-BB, an IC$_{50}$ of 894 pM towards VEGFA-121 and an IC$_{50}$ of 23 pM towards VEGFA-165.

As affinities in the low-pM range towards both VEGF isoforms, as well as PDGF-BB were desired, both paratopes in clone VP were subjected to affinity maturation. To this end, the bispecific VP sequence was re-diversified independently within the L-side paratope, resulting in an L-side affinity maturation phage display library that was selected against VEGFA-121, and within the H-side paratope, resulting in an H-side affinity maturation phage display library that was selected against PDGF-BB. In the construction of the affinity maturation libraries, some framework mutations were reverted back to VH3 and Vk1 germline, while the VH or Vk outer loops were allowed to deviate from germline, to allow selection of enhancing residues that may potentially

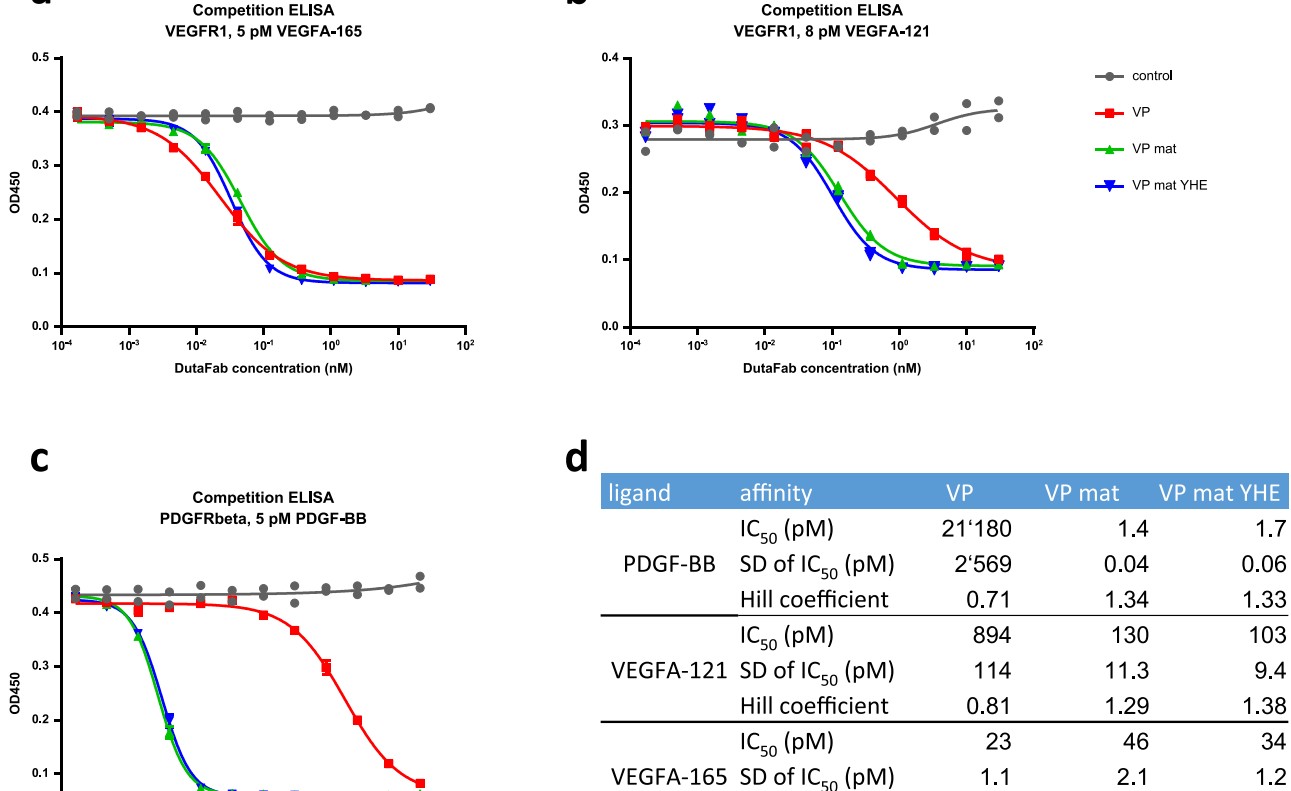

**Fig. 3 Affinity maturation of the VEGF-PDGF DutaFab.** The potency of the original bispecific DutaFab VP was significantly increased towards both PDGF-BB and VEGFA-121 by affinity maturation of both paratopes. Affinity maturation of either paratope did not negatively impact potency on the opposite paratope. **a** Competition ELISA with VEGFR1-Fc and 5 pM VEGFA-165. **b** Competition ELISA with VEGFR1-Fc and 8 pM VEGFA-121. **c** Competition ELISA with PDGFRβ-Fc and 5 pM PDGF-BB. **d** Tabulated curve fit parameters. SD = standard deviation. A sample size of $n = 2$ independently prepared samples was used.

modulate conformation of HCDR1 or LCDR1, respectively. Following independent affinity maturation of the L-side and H-side paratopes, the most potent paratope sequences were combined into a matured bispecific DutaFab (clone VP mat), with a variable region amino acid sequence as shown in Supplementary Fig. S1. As illustrated in Fig. 3a, a more than 10,000-fold increase in PDGF-BB potency was obtained, improving the $IC_{50}$ from 21 nM to 1.4 pM. In addition, a 7-fold increase in VEGFA-121 potency was obtained, improving the $IC_{50}$ from 894 pM to 130 pM (Fig. 3b), while the already high potency towards VEGFA-165 remained similar (Fig. 3c).

**Structure determination**. In order to confirm and further analyze the binding mode of DutaFabs, the affinity-matured DutaFab clone VP mat was subjected to X-ray crystallography, in complex with either VEGFA-121 or with PDGF-BB (Fig. 4 and Supplementary Table S1). A co-crystallization of DutaFab clone VP with both ligands bound simultaneously did not seem feasible due to the homodimeric nature of both ligands, potentially leading to multimeric complex formation and precipitation under the high concentrations needed for crystallography. Superposition of both complex structures via the Fab portion demonstrates that there is enough space for co-binding of both targets to the same DutaFab molecule (Fig. 4c). Experimentally, high-affinity co-binding of both targets to the same DutaFab molecule was confirmed by surface plasmon resonance (SPR, Fig. 4d).

As intended in the DutaFab concept, all 3 CDRs designed to be involved in H-side paratopes make contact with PDGF-BB and all 3 CDRs designed to be involved in L-side paratopes make contact with VEGFA-121 (Supplementary Fig. S1). Calculating solvent accessible surface areas both in presence and absence of the respective antigens yielded a water-excluded paratope size of approximately 430 $Å^2$ for the VEGFA-121 dimer on the L-side paratope and approximately 292 $Å^2$ of the PDGF-BB dimer on the H-side paratope. These values are in the same range as well-established therapeutic antibodies such as ranibizumab bound to VEGF (PDB 1CZ8), trastuzumab bound to Her2 (PDB 1N8Z), pertuzumab bound to Her2 (PDB 1S78) and atezolizumab bound to PD-L1 (PDB 5X8L) which show paratope sizes of 541 $Å^2$, 321 $Å^2$, 479 $Å^2$, and 493 $Å^2$, respectively.

**Further VEGF potency improvement based on structure**. VEGFA is a homodimer, which signals through VEGF receptors 1 and 2 by bringing two copies of the receptors together[31,32]. Signaling through the VEGF receptors is inhibited when one of the two receptor-binding sites on the VEGFA homodimer is blocked by an anti-VEGFA drug; however, single-blocked VEGF dimers retain the ability to bind to VEGF receptors on the cell surface in a monovalent manner. Single-blocked cell-bound VEGF dimers would allow rapid signaling via receptor molecules already present nearby in the cell membrane upon dissociation of the single drug copy. Only double-blocked VEGF dimers pose no such risk with regard to receptor signaling. For currently available anti-VEGF therapies, such double blocking is achieved by two copies of a Fab molecule such as ranibizumab, or one copy of a VEGF Trap such as aflibercept bound to the VEGF dimer.

Strikingly, examination of the co-crystal structure of DutaFab VP mat in complex with VEGF (Fig. 4b) revealed a close proximity of the DutaFab molecules when two copies of the DutaFab are bound to a VEGFA-121 homodimer. This observation prompted us to engineer the interface between the two VEGF-bound Fabs to make the simultaneous binding of two Fab copies to each VEGF dimer more energetically favorable, thus resulting in co-operative double-blocking of VEGF dimers. We

generated variants of DutaFab VP mat, with various point mutations in light chain positions Vk3, Vk5, Vk7, Vk100, and Vk104 which were identified as the contact between two Fabs bound to VEGF (Fig. 5a), and tested them in the potency assays shown in Fig. 3. In addition, a test was designed that examines VEGF-drug complex stability in the presence of competing VEGF receptor (VEGF baseline assay, Fig. 5b). In the baseline assay, an invariant VEGF concentration significantly greater than the drug $K_D$, in this case 1 nM, is pre-incubated with drug in a range of different concentrations, to form double-blocked VEGF-drug complexes with different excess amounts of drug. This double-blocked VEGF-drug mixture is then exposed to the high-affinity VEGF receptor VEGF-R1, coated on an ELISA plate at high concentration. Following incubation, the fraction of single-blocked and naked VEGF dimers that have lost the inhibitory drug and have become exposed on at least one receptor binding site, and have thus bound to the receptor on the plate, is detected using a polyclonal biotinylated anti-VEGF antibody. A co-operativity variant of DutaFab VP mat comprising the mutations Q3Y, T5H, S7E, Q100H, and L104V (clone VP mat YHE, Fig. 5a) displayed a significant increase in VEGF potency. In the VEGF $IC_{50}$ ELISAs with both VEGF isoforms, DutaFab VP mat YHE displayed a small improvement in $IC_{50}$ and Hill coefficient. Strikingly however, we observed a strong improvement in VEGF complex stability for the YHE variant as shown in Fig. 5b. On isoform VEGFA-121, the co-operative clone VP mat YHE was similarly potent to late-stage clinical reference molecules aflibercept and brolucizumab, while on isoform VEGFA-165 it was much more potent than the VEGF Trap molecule aflibercept. In a HUVEC proliferation assay and in a HEK293-NFAT reporter gene assay, the co-operativity engineering resulted in improved $IC_{50}$ and earlier complete inhibition of VEGF signaling at low drug concentrations, with significantly higher Hill coefficient (Fig. 5c). We measured the affinities of the further engineered DutaFab clone VP mat YHE for both VEGFA-121 and PDGF-BB, using the Kinetic Exclusion Assay (KinExA) method, which provides 1:1 interaction data determined entirely in solution and is thus suited to the dimeric nature of the targets that could potentially produce avidity effects in other experimental set-ups. As shown in Supplement Figs. S3 and S4, clone VP mat YHE has a high affinity for VEGFA-121 of $KD = 3$ pM, and a high affinity for PDGF-BB of $KD = 4$ pM. No signs for unspecific or polyreactive binding were observed for either DutaFab VP mat or DutaFab VP mat YHE (Supplementary Table S4).

**In vivo efficacy of the VEGF-PDGF DutaFab**. In order to examine the in vivo functionality of the VEGF-PDGF DutaFab, we tested the clone VP mat YHE in a laser-induced choroidal neovascularization (LCNV) model in brown Norway rats. Eyes were treated either 3 days before or on the day of laser induction using a single intravitreal administration of either DutaFab VP mat YHE, an anti-digoxigenin Fab fragment as negative isotype control[33], or the Fab fragment of the anti-VEGF antibody B20.4.1 as positive control[34]. Efficacy was assessed by fluorescence angiography 6 days after laser induction. The results suggest that treatment with the VEGF-PDGF DutaFab may provide additional efficacy compared with anti-VEGF monotherapy (Fig. 6). Given that the Fab fragments were dosed at a very high vitreal concentration of approximately 14 μM, expected to result in full neutralization of VEGF in case of the positive control, or both VEGF and PDGF in case of the DutaFab, this effect may be attributable to the dual biology of inhibiting both VEGF and PDGF. However, additional effects of the DutaFab's high VEGF potency and high folding stability cannot be excluded and could

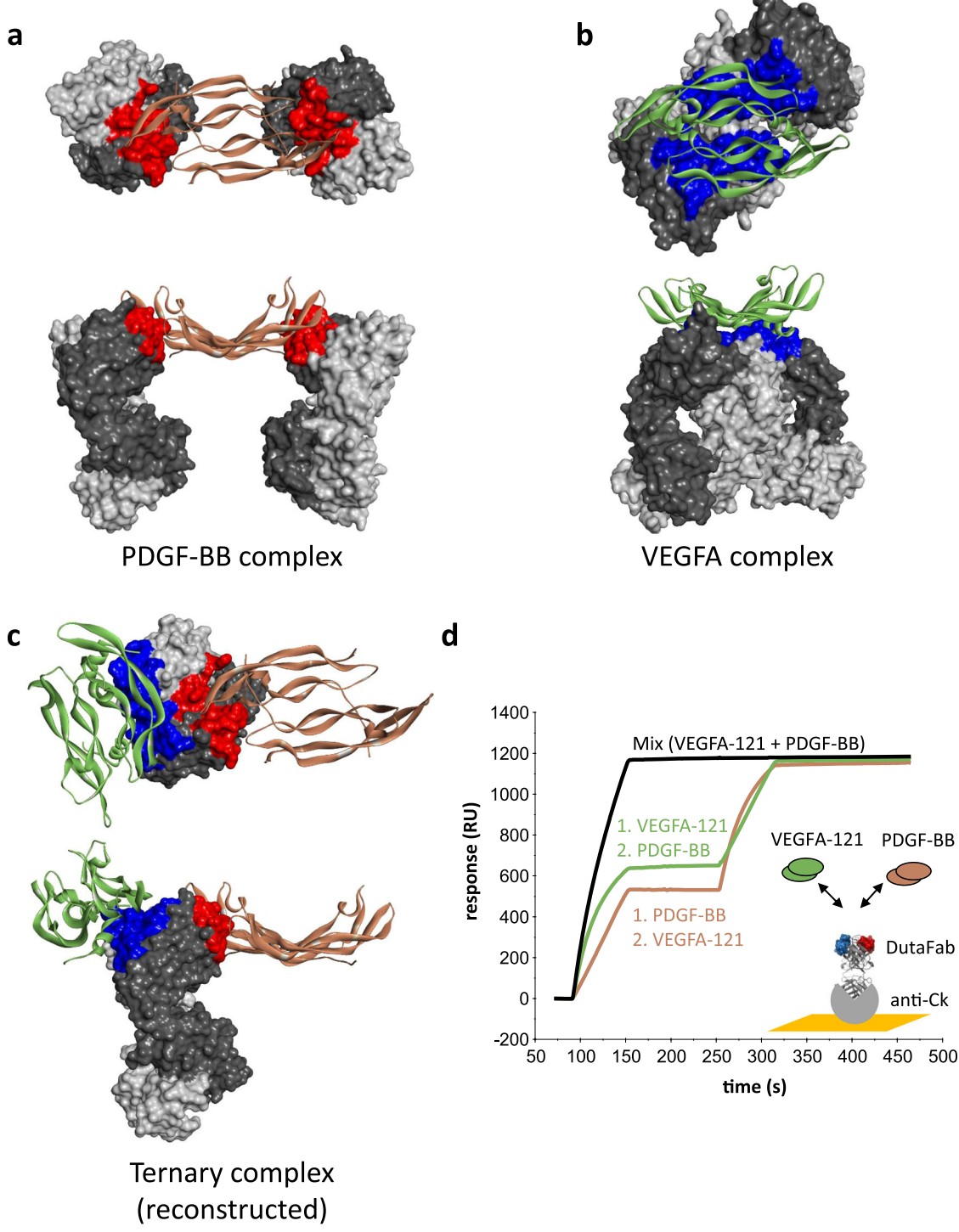

**Fig. 4 Structure analysis and co-binding. a** X-ray structure of the VP mat DutaFab in complex with PDGF-BB dimer. The Fab is shown in dark gray (heavy chain) and light gray (light chain). The PDGF-BB dimer (beige) binds to the H-side of the DutaFab with the interacting paratope depicted as red surface. The Fab-target complex is always shown in a view from the top and a view from the side. **b** X-ray structure of the VP mat DutaFab in complex with VEGFA dimer. The VEGFA dimer (green) binds to the L-side of the DutaFab with the interacting paratope depicted as blue surface. **c** Model of the ternary complex of the VEGF-PDGF DutaFab bound to its two targets simultaneously. The model was constructed by superpositioning the individual structures (**a**) and (**b**) via their Fab domains. **d** Concomitant binding demonstrated by SPR.

be addressed in future experiments. The study also revealed that DutaFab VP mat YHE had a normal ocular half-life of approximately 1.3 days in the retina, as expected for rat eyes, and normal retinal penetration with the expected concentration ratio in the retina and vitreous, thus confirming favorable stability and properties in vivo.

**Developability of VEGF-PDGF DutaFab.** DutaFabs have an inherent advantage for ophthalmology applications where the maximum feasible dose is limited by a small maximal injection volume, because their small size as compared to full-length bispecific antibodies, and their additional binding site as compared to conventional Fab fragments, provides higher valency per mass.

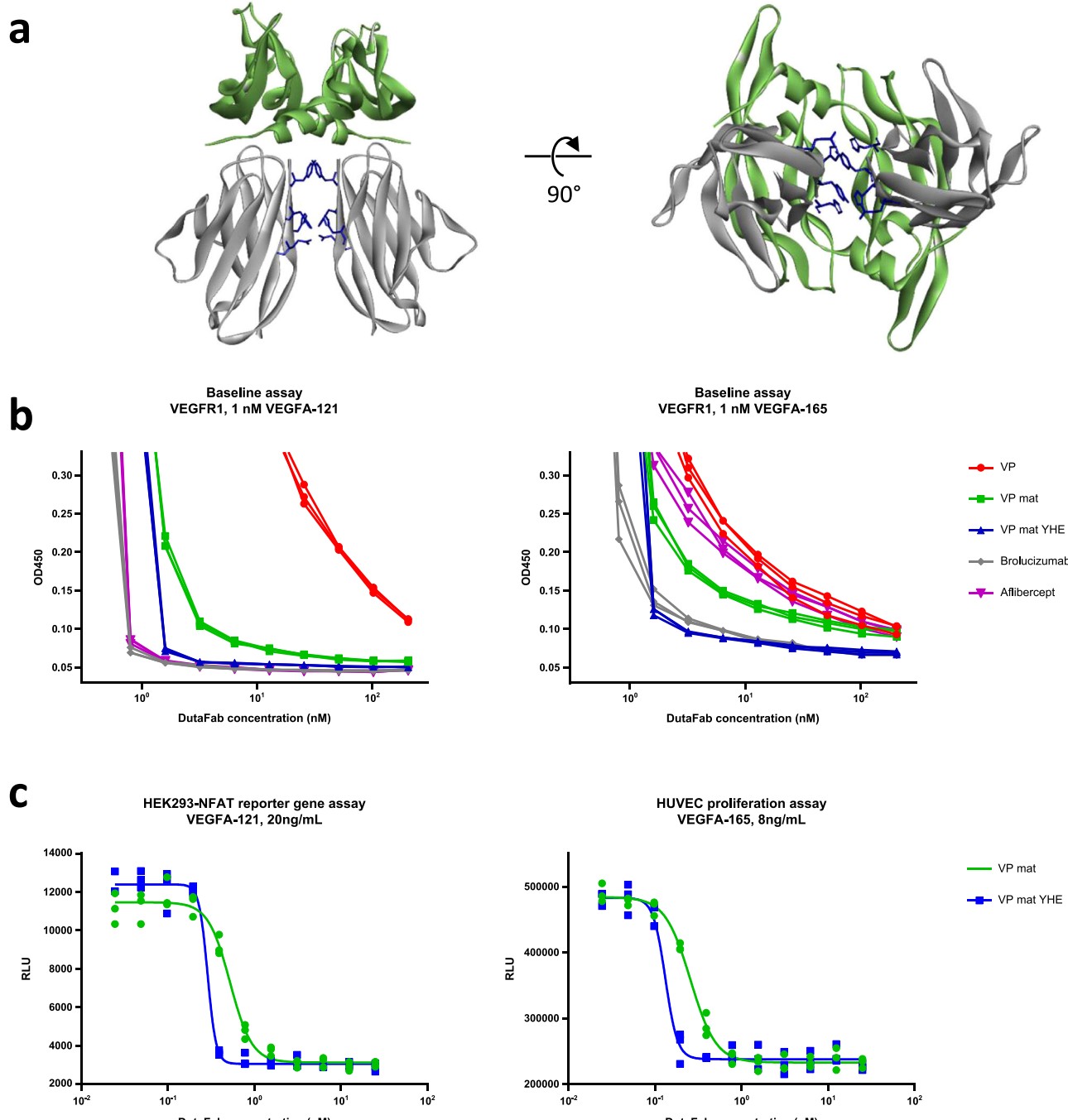

**Fig. 5 Structure-based engineering of VEGF potency. a** Side-view and bottom-view of the YHE co-operativity motif engineered into the Vk domain of the VP mat DutaFab. For clarity only the Vk domains (gray) are presented in complex with VEGF dimer (green). Residues involved in co-operative Fab binding are shown as blue sticks. Positions of sidechains have been modeled in silico using the Discovery Studio modeling suite based on the crystal structure of a DutaFab containing the co-operativity motif that is not presented in this study. **b** VEGF baseline assay with both VEGFA-121 and VEGFA-165 isoforms, showing complex stability of the VEGF dimer, fully blocked on both receptor binding epitopes by drug molecule. A low signal indicates complete blocking of the VEGF dimer. A high signal far away from baseline shows that a large fraction of VEGF has lost the blocking drug during the incubation with the competing receptor. The signal inflection seen at 1 nM drug paratope concentration is due to stoichiometric limitation. **c** HEK293-NFAT reporter gene assay and HUVEC proliferation assay with and without co-operativity motif. A sample size of $n = 3$ independently prepared samples was used in the experiments shown in panels **b** and **c**.

However, solubility and long-term stability are equally crucial for achieving a long durability of target suppression in the eye. We measured the viscosity of the most potent DutaFab clone VP mat YHE at high concentrations (Fig. 7a). While for intravitreal injections a viscosity of up to 20 cP is acceptable, clone VP mat YHE shows a significantly lower viscosity at the desired concentration of 200 mg/mL. We also measured the long-term stability at this high concentration for clone VP mat YHE, as relative aggregate concentration determined by SEC-HPLC. As shown in Fig. 7b, the clone shows no detectable increase in aggregate

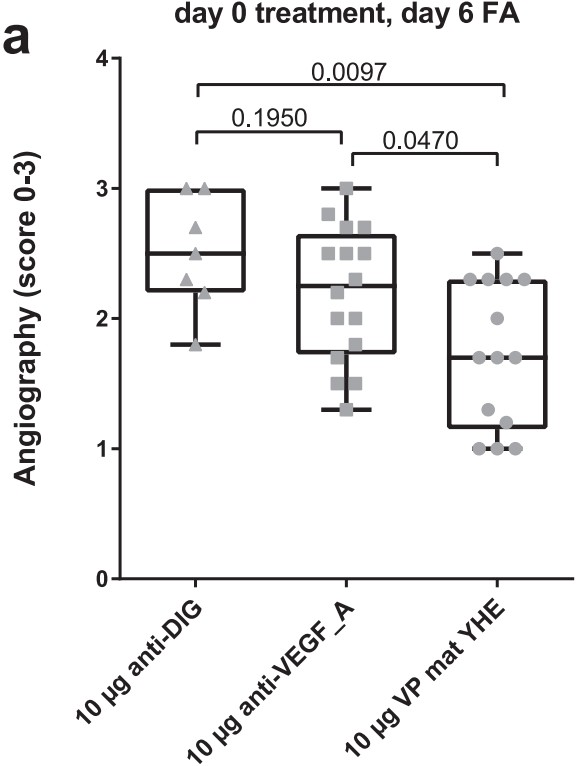

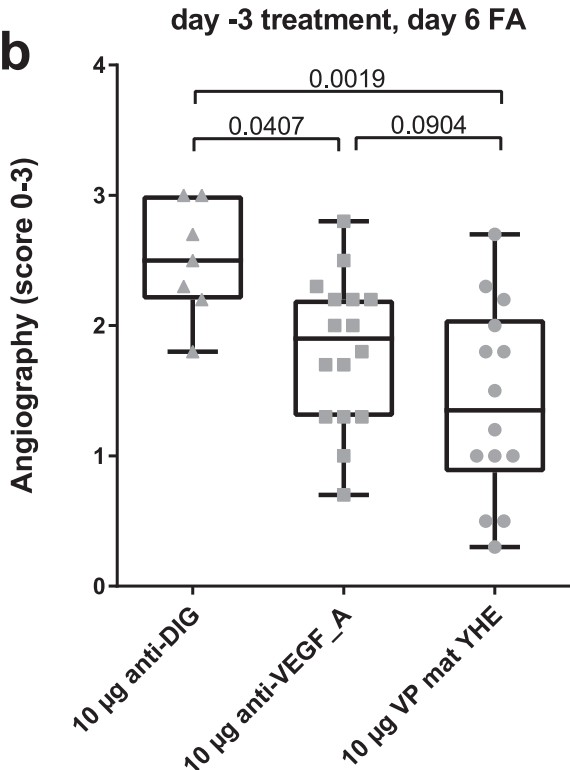

**Fig. 6 In vivo efficacy of the VEGF-PDGF DutaFab.** The in vivo efficacy of the most potent bispecific DutaFab against VEGFA and PDGF-BB, clone VP mat YHE, was analyzed using a laser-induced choroidal neovascularization (LCNV) model performed in pigmented Brown Norway rats. Each data point shown represents the average fluorescence angiography (FA) score for the lesions in one eye 6 days post laser treatment. **a** Result for antibody treatment by a single intravitreal administration on the day of laser induction. **b** Result for antibody treatment by a single intravitreal administration three days before laser induction. Sample size was $n = 7$ for anti-DIG, $n = 16$ for anti-VEGF_A and $n = 14$ for VP mat YHE. The whiskers represent the range from the lowest to the highest value in the set; the horizontal line within each box represents the median; boxes range from the 25th to the 75th percentile. Numbers over brackets represent adjusted p-values obtained from one-way ANOVA followed by the Holm–Šídák multiple comparison method.

## Discussion

We discovered that the human antibody Fv region can effectively be divided into two functional paratopes, comprising HCDR1, HCDR3, and LCDR2 on the so-called H-side and LCDR1, LCDR3, and HCDR2 on the L-side of the Fv. The achievable spatial separation of these two paratopes within the CDRs of the human Fv region represents a paradigm shift as compared to all prior bispecific approaches, because it allows simultaneous binding of two target molecules to the same Fv (Fig. 4) and thereby enables special therapeutic mechanisms.

First, the DutaFab binding mode effectively doubles the molar dose that can be administered per weight of therapeutic protein injected. In conventional Fab fragments, one target molecule can be neutralized per 50 kD of drug, while in DutaFabs one target molecule can be neutralized per 25 kD. This characteristic is especially interesting when combined with high solubility, and for the VEGF–PDGF DutaFab described here, we reached a solubility of >200 mg/mL with low viscosity, thus potentially enabling a high molar dose.

Second, the possibility of co-binding two targets or epitopes on the same Fv of a regular human Fab fragment also opens therapeutic options outside of the field of ophthalmology. The two paratopes on the Fv are in very close proximity with individual CDR residue distances in the range of 20–40 Å, thus enabling mechanisms of action that rely for example on clustering receptors or on bringing two proteins into functional distance to each other with potential applications in the fields of inflammation, neurology and oncology.

We observed that in the DutaFab generation process it is informative to combine a panel of L-side binders against a first target with a panel of H-side binders against a second target, rather than generating only a single paratope combination. When generating such a matrix, one may find that not all bispecific DutaFab clones exhibit simultaneous binding of the two target molecules to the same Fv, but that a sub-set of bispecifics exhibits mutually exclusive binding of the two target molecules, presumably due to steric hindrance between the two targets that depends on both epitope and target size, thereby allowing the selected binding mode to be matched to the envisaged therapeutic mechanism of action.

Of the numerous novel antibody formats and bispecific antibody technologies described over the years, many have not met the challenge of fulfilling all criteria important in developing genuinely usable antibody drugs that can succeed in the clinical setting. It has become clear that any novel features in bispecifics must not come at the expense of robust discoverability, developability, manufacturability and tolerability if they are to be truly beneficial.

species after 4 weeks at 4 °C at high concentration, and only a minor increase in aggregates when incubated for 4 weeks at 40 °C, indicating suitability for the development of high-concentration liquid formulations. Moreover, a late aggregation onset temperature of approximately 70 °C for DutaFab VP mat YHE was determined by static light scattering (Fig. 7c).

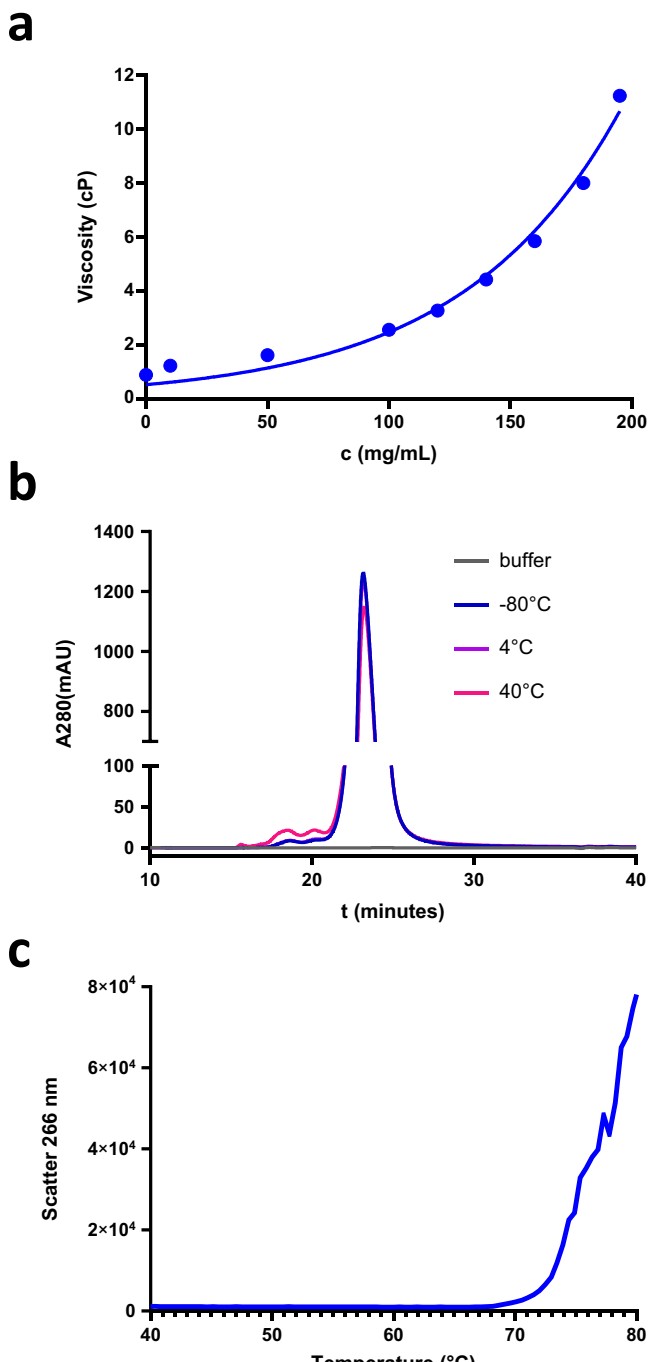

**Fig. 7 CMC and biophysical properties of DutaFabs. a** Viscosity of clone VP mat YHE (25 °C, 20 mM His pH 6.0). **b** Long-term RAC after PBS incubation, measured by SEC-HPLC after 4 weeks storage of clone VP mat YHE at 200 mg/mL. **c** Aggregation onset analysis of DutaFab clone VP mat YHE, 1 mg/mL, 20 mM His pH 6.0.

In this work, we show that the 3-CDR paratopes created in the Fv region of DutaFabs are genuinely structurally independent, as they can be mixed and matched or separately optimized in a modular manner. We discovered that an H-side and an L-side paratope obtained as monospecific binders can be merged into bispecific DutaFabs with a high success rate; the affinity and specificity of target binding were nearly always retained in the merged bispecific clones. We also demonstrate that one paratope can be affinity matured without loss of binding at the

opposite paratope. This modularity might be facilitated by the high thermostability of DutaFab scaffolds. Fab thermostabilities of up to 92 °C have been described by Demarest and colleagues[35], while some DutaFab scaffolds can reach Fv thermostabilities above 100 °C. Furthermore, we demonstrate that a DutaFab specific for VEGFA and PDGF-BB could be affinity matured to reach very high potency and affinity for both of these targets (Fig. 3, Supplementary Figs. S3 and S4), as required for demanding clinical applications in ophthalmology, without destroying the affinity or specificity of binding on the opposite side. The tight binding of the VEGF-PDGF DutaFab to each of its two targets was achieved by using only three CDRs per target. This demonstrates that by offering the appropriate diversity when designing libraries, paratope sizes and more importantly, paratope affinities sufficient for a challenging clinical scenario can be achieved using only half the CDRs of conventional antibodies. Finally, the most potent VEGF–PDGF DutaFab clone shows promising developability with excellent physico-chemical and manufacturing properties, comparable to the most modern conventional monospecific antibody fragments.

Humanness, i.e., the sequence conservation relative to the closest fully human variable domain, is considered to be an important factor in reducing the immunogenicity risk of antibody drugs. All three VEGF-PDGF DutaFab clones presented here deviate from the closest matching human germline frameworks only by a modest number of mutations, comparable to established antibodies in clinical development. Indeed, in a phylogenetic tree, all three DutaFabs show a close relationship within 133 late-stage clinical human and humanized antibodies and do not form a separate branch (Supplementary Fig. S2a). Moreover, with regard to main chain conformation, DutaFabs are structurally indistinguishable from conventional human Fab drugs, as illustrated by superposition of the structures of DutaFab VP mat YHE and ranibizumab (Supplementary Fig. S2b).

The present case study of a DutaFab specific for VEGFA and PDGF-BB is aimed at the field of ophthalmology, where the high solubility and simultaneous binding properties of the molecule allow for an exceptionally high molar dose of two very potent paratopes. The DutaFab shows potency towards two targets that is competitive with the most advanced monospecific antibody approaches. Its very high affinity for PDGF-BB is reflected in an $IC_{50}$ of approximately 2 pM (Fig. 3a), and its very high affinity for VEGFA isoforms 121 and 165 is reflected in an $IC_{50}$ values that are also in the low pM range (Fig. 3b, c). We further tested VEGF-drug complex stability in the presence of competing VEGF receptor, using a specially designed VEGF baseline assay (Fig. 5b). We were particularly interested in the performance of the DutaFab compared to late-stage clinical reference compounds on the VEGFA isoform 165, which comprises the additional heparin binding domain, because this is considered to be especially critical for the pathogenicity of VEGF[36]. Here the most potent DutaFab VP mat YHE showed excellent VEGF complex stability, being much more potent than the VEGF Trap molecule aflibercept, even though this has an inherent propensity towards double-blocking VEGF dimers based on avidity (Fig. 5b). Thus, by offering an exceptional combination of potency and solubility, the co-operative VEGF paratope developed in this work lends itself as a highly attractive component of future ophthalmology drugs.

In conclusion, with the DutaFab platform a class of dual targeting human Fab fragments has become available, which combine several distinct and useful features that make them ideally suited to the development as antibody therapeutics in the field of ophthalmology and beyond. Indeed, DutaFabs have reached the stage of clinical development; for example, clinical trial NCT04567303 investigates a DutaFab in patients with neovascular AMD.

## Methods

**Naïve library construction**. Phagemid vector pDuta4 was constructed from the pUC19 backbone (purchased from NEB) by addition of an M13 origin and a Fab display cassette under control of the lacZ promotor, comprising LC signal peptide, light chain, HC signal peptide, heavy chain, linker and domain 3 of the gene 3 protein from phage Fd. Diversified Fab raw libraries were generated by gene synthesis of the region encoding VH and Vk variable domains, and naïve raw libraries were amplified by PCR. Diversification was introduced using various chemistries for different libraries, including nucleotide-based diversification and trinucleotide muta-genesis (TRIM). For affinity maturation, DutaFab genes were re-diversified by overlap PCR amplification, using combinations of degenerate oligonucleotides. The re-diversification was always directed at one paratope, either L-side or H-side. Amplified naïve libraries or affinity maturation libraries were cloned into the pDuta4 vector, transformed into TG1 E. coli cells, and rescued using M13K07 helper phage according to standard molecular biology methods.

**Panning**. For DutaFab panning, various protein targets were expressed from synthetic DNA and purified, or sourced from commercial suppliers. In this study, targets were purchased from Peprotech (VEGFA-121, catalog number 100-20A; PDGF-BB, catalog number 100-14B). Targets were biotinylated using EZ-Link Sulfo-NHS-SS-Biotin (ThermoFisher catalog number A39258), according to manufacturer's instructions. For naïve selections, phage library panning was typically performed in 4 rounds, wherein the first round was performed with 100 nM of biotinylated target pre-immobilized on Dynabeads M-280 Streptavidin (Thermofisher catalog number 11206D), and rounds 2–4 were performed with 75 nM, 15 nM, and 3 nM of biotinylated target in solution, followed by capture of Fab-on-phage/target complexes on the M-280 beads. For affinity maturation, all panning rounds were performed with the biotinylated target in solution, using various target concentrations in different selection arms. Captured phage clones bearing target-specific DutaFabs were eluted from the M-280 beads using 100 mM DTT, used for infection of log-phase TG1 E. coli cells, and rescued using M13 K07 helper phage, according to standard protocols.

**Screening**. For screening of selection outputs, a polyclonal plasmid miniprep of the respective selection round was prepared from the infected TG1 E. coli cells. Plasmids were digested using BamHI restriction endonuclease, which cuts the phagemid pDuta4 upstream and downstream of the phage Fd gene 3 domain. Plasmids were re-circularized by ligation, generating an in-frame fusion of a T7 tag at the C-terminus of the DutaFab CH1 domain. The ligated polyclonal plasmids encoding T7-tagged DutaFabs were transformed into TG1 E. coli cells (Zymo Research catalog number T3017), and single colonies were picked into microtiter plates. Soluble DutaFabs were expressed in microtiter plates and supernatants were clarified by centrifugation. The DutaFab culture supernatants were screened using either a direct binding ELISA, a capture ELISA or a competition ELISA that, briefly, utilized the following assay steps. In the direct binding assay, targets were coated onto Nunc Maxisorp plates and exposed to DutaFab culture supernatants, and specifically bound clones were detected using a Novagen biotinylated anti-T7 antibody (Merck catalog number 69968). In the capture assay, a polyclonal anti-Fab antibody was coated onto Nunc Maxisorp plates, DutaFabs were captured from culture supernatants, biotinylated target was added in solution, and specifically bound biotinylated target was detected using KPL Streptavidin-HRP (Sera-care catalog number 5270-0029). In the competition assay, relevant target ligands (in this case VEGF-R1-Fc or PDGF-Rβ-Fc) were coated onto Nunc Maxisorp plates, a pre-incubated mixture of DutaFab culture supernatant and unlabeled VEGFA or PDGF-BB was added, and free target not neutralized by DutaFab and able to bind to its receptor was detected using a biotinylated anti-VEGFA or anti-PDGF-BB antibody, respectively. DutaFab clones giving the highest direct binding or capture assay signals or the most complete inhibition of target binding to the receptors were analyzed by DNA sequencing.

**Mid-scale expression and purification**. DutaFabs were expressed either by periplasmatic expression in E. coli using standard procedures and vectors. Alternatively, DutaFabs were expressed in HEK Expi293 cells (ThermoFisher). Purification was accomplished by a capture step on kappa select resin (GE healthcare), followed either by a cation exchange chromatography step on PorosA (Thermo-Fisher) or a preparative size-exclusion chromatography using a Superose12 column (GE Healthcare).

**Surface-plasmon resonance to demonstrate concomitant binding**. Approximately 5000 resonance units (RU) of the capturing system (CaptureSelect™ Human Fab kappa Kinetics Biotin Conjugate (Thermo Fischer scientific)) were coupled on a CM5 chip (GE Healthcare) at pH 5.0 by using an amine coupling kit (GE Healthcare). The sample and system buffer was PBS-T (1 mM $KH_2PO_4$, 10 mM $Na_2HPO_4$, 137 mM NaCl, 2.7 mM KCl pH 7.4, 0.05% Tween20, Roche Diagnostics GmbH). All measurements were performed at 25 °C. The DutaFab was captured by injecting a 5 µg/ml solution for 60 s at a flow rate of 5 µl/min. Independent binding of each ligand to the DutaFab was analyzed by determining the active binding capacity for each ligand, either added sequentially or simultaneously at a flow rate of 5 µl/min. Concentrations of 1.25 µg/mL PDGF-BB and 5 µg/mL of VEGFA-121 were used, respectively. The surface was regenerated by injection of 10 mM Glycine

pH 2.0 for 90 s at a flow rate of 30 µl/min. Bulk refractive index differences were corrected for by subtracting the response obtained from the reference flow cell without captured DutaFab.

**Affinity determination by kinetic exclusion assay**. Affinity determination was performed by Kinetic Exclusion Assay, using a Sapidyne Instruments KinExA instrument. All data were based on duplicate titration series, performed as 1:2 dilutions. For affinity determination of DutaFab towards VEGFA-121, a 4 h pre-incubation was performed of 100 pM DutaFab and a titration of 122 fM–2 nM VEGFA-121 ligand. For affinity determination of DutaFab towards PDGF-BB, a 4 h pre-incubation was performed of 50 pM DutaFab and a titration of 244 fM–4 nM PDGF-BB ligand. The calculation of KD values and 95% confidence intervals was performed using Sapidyne Instruments software, according to manufacturer's instructions.

**IC50 and baseline ELISA**. 96-well plates (Maxisorp, NUNC) were coated with 50 µl/well containing 1 µg/mL rhVEGFR-1-Fc or 0.5 µg/mL rhPDGFRβ-Fc (both from R&D Systems) in 200 mM $NaHCO_3$, pH 9.4 (Thermo Fisher) and incubated at RT for 1 h. In a 96-well round bottom PS plate, ligand-drug mixtures were prepared for 1.5 h pre-incubation with the same ligand concentration in all wells and a 1:2 serial dilutions of the Fabs/Mabs in PBS supplemented with 0.1% Tween-20 and 1% BSA (Sigma-Aldrich) in columns 1–11 with the highest concentration in column one and with no Fab or Mab in column 12. The following ligands were tested: recombinant human VEGFA-121, recombinant human VEGFA-165 (both from Humanzyme) and recombinant human PDGF-BB (R&D Systems). The coated Maxisorp plates were washed 2x with PBST and blocked by filling the wells with PBST supplemented with 2% (w/v) skimmed milk powder (Carl Roth, T145.3) and incubating 45 min at RT. After washing the Maxisorp plates 2× with PBST, 50 µL/well from the ligand-drug plate were transferred to the Maxisorp plate and incubated for 10 min (IC50 ELISA) or 1.5 h (Baseline ELISA) at RT. The plates were washed 2× with PBST and 50 µL/well of 0.1 µg/mL biotinylated anti-hVEGF or anti-hPDGF-BB goat IgG (R&D Systems) and 0.25 µg/ml Peroxidase-labeled Streptavidin (KPL) was added for detection. After a 30 min incubation at RT, plates were washed 6× with PBST and developed by the addition of 50 µL/well colorimetric TMB substrate (Sera Care). The reaction was stopped by addition of 50 µL/well of 1 N sulfuric acid and absorption was measured at 450 nm. Synthetic genes encoding the reference compounds aflibercept and brolucizumab were purchased according to the published INN sequences, cloned and expressed in HEK293 cells. IC50 values and Hill coefficients were calculated with a 4-parameter fit method using GraphPrism software.

**Cell-based VEGF assays**. For the NFAT reporter gene assay, GloResponse™ NFAT-RE-luc2P HEK293 cells (Promega) were seeded at a density of 40,000 cells per well in FreeStyle medium (Gibco). DutaFabs were pre-incubated with VEGF (20 ng/mL), incubated for 30 min and added to the cell suspension. After 5 h incubation at 37 °C, BioGlo reagent (Promega) was added according to the manufacturer's instruction and luminescence was read using an Infinite Pro microplate reader (Tecan). For the HUVEC proliferation assay, human umbilical vein/vascular endothelial cells (HUVEC) (ATCC) were seeded on collagen-coated 96-well plates (ACEA Biosciences) at a density of 3000 cells per well and incubated in starvation medium (Promocell) for 24 h. DutaFabs were pre-incubated with VEGFA-165 (8 ng/mL), incubated for 30 min and added to the cell suspension. Proliferation was quantified after 48 h with the CellTiter-Glo® system (Promega) according to the manufacturer's instructions using an Infinite Pro microplate reader (Tecan).

**Crystallization of the DutaFab-VEGF complex**. To obtain crystals of the DutaFab in complex with VEGFA-121 (Peprotech catalog number 100-20A) both proteins were mixed at a 1:1 molar ratio in relation to VEGF monomer. The complex was concentrated to 11 mg/ml and crystallization was performed by hanging drop vapor diffusion against 0.1 M MES pH 6.5 and 1.6 M magnesium sulfate at 20 °C. Needle-shaped crystals grew in about 120 days and were frozen in liquid nitrogen with 20% glycerol as cryo-protectant.

**Crystallization of the DutaFab-PDGF Complex**. To obtain crystals of the Duta-Fab in complex with PDGF-BB (Peprotech catalog number 100-13A) both proteins were mixed at a 1:1 molar ratio in relation to PDGF monomer. The complex was concentrated to 12 mg/mL and crystallization was performed by hanging drop vapor diffusion against 0.1 M Tris pH 7.5, 42% (+/−)-2-methyl-2,4-pentanediol (MPD) at 20 °C. Plate-shaped crystals grew within a few weeks and were frozen in liquid nitrogen with 20% glycerol as cryo-protectant.

**Data collection and crystal structure determination DutaFab-VEGF Complex**. Data were collected at a temperature of 100 K at station I911-3 at the MAX IV Laboratory, Lund, Sweden, equipped with a MarMosaic 225 detector. A total of 200 diffraction images were collected with an exposure time of 30 s and an oscillation range of 1° per image. The data were integrated and scaled using XDS[37], then merged and converted to the MTZ format using XDSCONV. The CCP4 suite[38] was used to solve and refine the structure of the DutaFab:VEGF complex via molecular

replacement using Phaser[39]. The solvent content and Matthews' coefficient were calculated to be 43.5% and 2.18 Å³/Da, respectively, which corresponds to two DutaFab:VEGF complexes in the asymmetric unit As search models, the structure of VEGF (PDB 1BJ1) and a related Fab (PDB 1JPS) were used to search for one and two copies in the asymmetric unit, respectively. Phaser could localize the dimeric VEGF and the two Fab molecules. After molecular replacement, rigid body refinement was done in Refmac5[40]. Iterative restraint refinement in Refmac5 and model building in Coot[41] resulted in a final structure with R and $R_{free}$ values of 22.2% and 27.2%, respectively. The final structure was deposited to the PDB with the accession code 6T9D.

**Data collection and crystal structure determination DutaFab-PDGF Complex.**
Data were collected at a temperature of 100 K at station I03 of the Diamond Light Source, UK equipped with a Pilatus3 6 M detector. A total of 800 diffraction images were collected with an exposure time of 0.1 s and an oscillation range of 0.2° per image. The data were integrated and scaled using XDS[37], then merged and converted to the MTZ format using XDSCONV.

The CCP4 suite[38] was used to solve and refine the structure of the DutaFab: PDGF-BB complex via molecular replacement using Phaser[39]. The solvent content and Matthews' coefficient were calculated to be 53.2% and 2.63 Å³/Da, respectively, which corresponds to two DutaFab:PDGF complexes in the asymmetric unit. As search models the monomeric PDGF (PDB 4QCI) and the Fab from the DutaFab: VEGF structure were used to search for two copies each in the asymmetric unit. Phaser could localize the two Fab molecules. After molecular replacement, rigid body refinement was done in Refmac5[40]. The first round of restrained refinement resulted in a map clear enough that the dimeric PDGF-BB could be manually placed. Iterative restrained refinement in Refmac5 and model building in Coot[41] resulted in a final structure with R and $R_{free}$ values of 26.7% and 30.9%, respectively. The final structure was deposited to the PDB with the accession code 6T9E.

**Laser-induced CNV model.** The laser-induced choroidal neovascularization (LCNV) study was performed in pigmented Brown Norway rats, which were males and 8-10 weeks old at the time of induction. Only animals with no visible defect were randomly assigned to the study groups. Animals were housed in standard cages (two or three animals per cage), under identical environmental conditions at a temperature of 22 ± 2 °C and 55 ± 10% relative humidity. Animals were routinely exposed (in-cage) to 10–200 lx light in a 12-h light and darkness cycle. Neovascularization was induced in six positions in both eyes by applying 170 mW of 532-nm laser light on 75-ʃm spots around the optic nerve, between the main retinal vessel branches, through the slit lamp and a contact lens. Test compounds anti-Dig[33] (negative control, used for 7 eyes), anti-VEGFA B20.4.1 (positive control, used for 16 eyes[34]) or DutaFab VP mat YHE (bispecific study molecule, used for 14 eyes) were used in the Fab fragment format, and their affinities towards rat VEGF are listed in Supplementary Table S2. The compounds were administered intravitreally in both eyes as a single 10 ʃg dose in a 3 ʃL injection either on day −3 or on day 0, in two study groups, using a 30-G-needle. Based on the rat vitreous volume of up to 14 μL and Fab molecular weight of 50 kDa, this dose results in a vitreal concentration of approximately 14 μM Fab, several 1000-fold above the KDs of the antibodies and expected to fully neutralize the rat VEGF and/or PDGF targets. On day 6 after induction, fluorescein angiography was performed using 250 ʃL per 100 g body weight of 10% sodium fluorescein and a Heidelberg Retinal Angiograph. The leakage of fluorescein was evaluated in the angiograms by two masked examiners to the study groups and graded as follows: Score 0 = no leakage, Score 1 = slightly stained, Score 2 = moderately stained and Score 3 = strongly stained. If the two scores assigned to a lesion did not match, the higher score was used for analysis. All standard operating procedures and protocols described in this manuscript have been reviewed by Iris Pharma Internal Ethics Committee. All animals were treated according to the Directive 2010/63/UE European Convention for the Protection of Vertebrate Animals used for Experimental and Other Scientific Purposes and to the Association for Research in Vision and Ophthalmology (ARVO) Statement for the Use of Animals in Ophthalmic and Vision Research.

**Viscosity.** Viscosity measurement was essentially performed as described by He et al.[42]. Briefly, samples were concentrated to various protein concentrations in 20 mM histidine/histidine chloride pH 6.0, before polystyrene latex beads (300 nm diameter) and Polysorbate 20 (0.02% v/v) were added. Samples were transferred into an optical 384-well plate by centrifugation through a 0.4 μm filter plate and covered with silicone oil. The apparent diameter of the latex beads was determined by dynamic light scattering at 25 °C. The viscosity of the solution can be calculated as $\eta = \eta_0(r_h/r_{h,0})$ ($\eta$: viscosity; $\eta_0$: viscosity of water; $r_h$: apparent hydrodynamic radius of the latex beads; $r_{h,0}$: hydrodynamic radius of the latex beads in water).

**Stress test and SEC.** Aggregation stability at high concentration was determined by quiescent incubation of samples for 4 weeks at a concentration of 200 mg/mL in 20 mM His/His-HCl, 160 mM Sucrose, 0.02% Poloxamer 188, pH 5.5 at 40 °C, 4 °C, and <−60 °C, respectively. Aggregate levels were determined by size-exclusion chromatography using a Superdex 75 Increase 10/300 GL column (GE Healthcare) with PBS pH 7.4 as the mobile phase.

**Differential scanning calorimetry (DSC).** The thermostability of Fab fragments was determined by DSC in 1× PBS pH 7.4 (Invitrogen catalog No. 10010056), using a capillary cell microcalorimeter and VPViewer2000 CapDSC software (MicroCal). Parameters were set to scanning a temperature window of 32 °C to between 105 °C and 115 °C, a pre-scan thermostat of 2 min, a post-scan thermostat of 0 min, no gain, and scan rates of 1 °C per minute or 4 °C per minute. Melting temperatures of Fab fragments were determined after PBS reference subtraction, using Origin 7.0 software. Data were fitted to the Non-2-State model, using 200 Levenberg-Marquardt iterations.

**Aggregation onset.** Samples were prepared at a concentration of 1 mg/mL in 20 mM histidine/histidine chloride pH 6.0, transferred into a 10 μL micro-cuvette array and static light scattering data upon irradiation with a 266 nm laser were recorded with an Optim1000 instrument (Avacta Inc.), while they were heated at a rate of 0.1 °C/min from 25 °C to 90 °C.

**Reporting summary.** Further information on research design is available in the Nature Research Reporting Summary linked to this article.

## Data availability
X-ray crystallography data that support the findings of this study have been deposited in the Protein Database (www.rcsb.org) with the accession codes 6T9D and 6T9E, respectively. Calorimetry raw data shown in Fig. 2a, SPR data shown in Fig. 4d, stability and viscosity data shown in Fig. 7, and Kinexa binding instrument readouts shown in Supplementary Figs. S3 and S4 are available from the corresponding author upon reasonable request. The authors declare that all other data supporting the findings of this study are available within the paper and its supplementary information files. Source data are provided with this paper.

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

## Author contributions

R.B. and K.J. designed the libraries and developed the panning and affinity maturation strategies. S.Fenn, J.S., K.K., A.M., and M.R. performed in-vitro lab experiments. D.T.L. and R.K. determined the X-ray structures. R.B., H.K., and S.Fenn evaluated the data and wrote the manuscript. M.S. and S.Fauser supervised this work.

## Competing interests

At the time of writing of this manuscript, the following authors were paid employees of Roche: R.B., K.J., S.Fenn, J.S., K.K., A.M., M.R., S.Fauser, M.S., H.K. D.T.L. and R.K. were paid employees of Saromics Biostructures AB.
