## [Peer Review File · Nature Communications]

Fig. 3a: VEGFA-165

DutaFab concentration (nM)	control		VP		VP mat		VP mat YHE	
30	0.406	0.408	0.087	0.091	0.088	0.089	0.086	0.084
10	0.391	0.403	0.087	0.088	0.087	0.087	0.083	0.084
3.333333333	0.394	0.394	0.089	0.092	0.088	0.086	0.082	0.083
1.111111111	0.393	0.403	0.094	0.094	0.088	0.09	0.084	0.086
0.37037037	0.386	0.395	0.107	0.108	0.097	0.099	0.086	0.087
0.12345679	0.383	0.394	0.132	0.134	0.14	0.14	0.107	0.108
0.041152263	0.388	0.401	0.194	0.212	0.246	0.254	0.214	0.217
0.013717421	0.385	0.397	0.279	0.281	0.327	0.338	0.328	0.326
0.004572474	0.397	0.39	0.33	0.336	0.361	0.366	0.367	0.367
0.001524158	0.391	0.393	0.366	0.375	0.378	0.367	0.381	0.387
0.000508053	0.387	0.4	0.382	0.384	0.377	0.376	0.389	0.378
0.000169351	0.392	0.4	0.387	0.403	0.397	0.398	0.398	0.401

Fig. 3b: VEGFA-121

DutaFab concentration (nM)	control		VP		VP mat		VP mat YHE	
30	0.337	0.312	0.101	0.101	0.093	0.094	0.091	0.091
10	0.333	0.293	0.106	0.112	0.093	0.106	0.09	0.089
3.333333333	0.313	0.293	0.141	0.136	0.091	0.091	0.086	0.09
1.111111111	0.292	0.285	0.19	0.185	0.094	0.096	0.088	0.087
0.37037037	0.287	0.277	0.228	0.225	0.135	0.138	0.111	0.107
0.12345679	0.268	0.27	0.271	0.268	0.203	0.211	0.195	0.192
0.04115226	0.283	0.262	0.288	0.268	0.26	0.262	0.249	0.245
0.01371742	0.297	0.268	0.283	0.287	0.301	0.291	0.289	0.29
0.00457247	0.274	0.275	0.308	0.297	0.292	0.306	0.31	0.298
0.00152416	0.286	0.288	0.3	0.296	0.308	0.317	0.325	0.303
0.00050805	0.294	0.295	0.31	0.301	0.304	0.331	0.31	0.317
0.00016935	0.291	0.262	0.293	0.299	0.291	0.298	0.283	0.29

Fig. 3c: PDGF-BB

DutaFab concentration (nM)	control		VP		VP mat		VP mat YHE	
1000	0.446	0.468	0.08	0.085	0.064	0.062	0.06	0.058
200	0.445	0.442	0.119	0.12	0.062	0.059	0.06	0.059
40	0.45	0.434	0.202	0.199	0.06	0.058	0.056	0.059
8	0.44	0.447	0.307	0.289	0.059	0.059	0.06	0.057
1.6	0.45	0.418	0.37	0.365	0.061	0.061	0.062	0.059
0.32	0.445	0.431	0.396	0.396	0.063	0.062	0.061	0.061
0.064	0.442	0.421	0.427	0.421	0.064	0.064	0.063	0.062
0.0128	0.451	0.428	0.413	0.422	0.074	0.073	0.074	0.074
0.00256	0.439	0.415	0.41	0.41	0.186	0.169	0.208	0.191
0.000512	0.444	0.421	0.397	0.41	0.357	0.356	0.365	0.358
0.000102	0.443	0.426	0.424	0.415	0.418	0.413	0.411	0.411
0.0000205	0.444	0.427	0.423	0.429	0.442	0.429	0.424	0.435

d

ligand	affinity	VP	VP mat	VP mat YHE
PDGF-BB	IC ₅₀ (pM)	21'180	1.4	1.7
	SD of IC ₅₀ (pM)	2'569	0.04	0.06
	Hill coefficient	0.71	1.34	1.33
VEGFA-121	IC ₅₀ (pM)	894	130	103
	SD of IC ₅₀ (pM)	114	11.3	9.4
	Hill coefficient	0.81	1.29	1.38
VEGFA-165	IC ₅₀ (pM)	23	46	34
	SD of IC ₅₀ (pM)	1.1	2.1	1.2
	Hill coefficient	0.9	1.4	1.6

Fig. 5b: VEGFA-121

DutaFab concentration (nM)	VP			VP mat			VP mat YHE			Brolucizumab			Aflibercept		
204.8	0.112	0.113	0.109	0.059	0.058	0.057	0.086*	0.05	0.051	0.047	0.046	0.048	0.047	0.046	0.047
102.4	0.154	0.151	0.147	0.058	0.058	0.059	0.051	0.051	0.05	0.046	0.045	0.045	0.044	0.046	0.044
51.2	0.207	0.205	0.203	0.06	0.062	0.062	0.051	0.052	0.051	0.046	0.046	0.045	0.045	0.046	0.045
25.6	0.288	0.272	0.263	0.066	0.065	0.067	0.053	0.053	0.053	0.046	0.047	0.046	0.045	0.046	0.046
12.8	0.382	0.37	0.384	0.075	0.071	0.074	0.054	0.054	0.054	0.047	0.047	0.047	0.046	0.047	0.047
6.4	0.534	0.522	0.555	0.081	0.084	0.085	0.055	0.055	0.056	0.048	0.049	0.049	0.048	0.05	0.049
3.2	0.774	0.767	0.764	0.104	0.107	0.11	0.057	0.057	0.057	0.05	0.053	0.051	0.051	0.053	0.052
1.6	1.001	0.965	0.994	0.208	0.208	0.221	0.071	0.075	0.074	0.056	0.057	0.056	0.058	0.059	0.058
0.8	1.166	1.122	1.111	0.654	0.642	0.622	0.527	0.517	0.572	0.069	0.076	0.069	0.081	0.086	0.087
0.4	1.312	1.275	1.278	1.034	1.052	1.042	1.086	1.111	1.114	0.627	0.679	0.567	1.002	1.096	1.048
0.2	1.385	1.334	1.366	1.278	1.275	1.297	1.315	1.319	1.273	1.065	1.15	1.115	1.34	1.39	1.282
0.1	1.468	1.397	1.49	1.425	1.409	1.464	1.416	1.42	1.39	1.368	1.399	1.364	1.369	1.42	1.397

*excluded due to instrument failure

Fig. 5b: VEGFA-165

DutaFab concentration (nM)	VP			VP mat			VP mat YHE			Brolucizumab			Aflibercept		
204.8	0.093	0.103	0.104	0.09	0.099	0.098	0.067	0.071	0.069	0.066	0.068	0.068	0.09	0.097	0.099
102.4	0.105	0.123	0.117	0.094	0.103	0.1	0.067	0.073	0.07	0.066	0.07	0.069	0.101	0.11	0.11
51.2	0.118	0.143	0.134	0.102	0.111	0.108	0.071	0.076	0.073	0.072	0.075	0.07	0.118	0.129	0.129
25.6	0.141	0.162	0.155	0.113	0.121	0.116	0.075	0.077	0.079	0.078	0.078	0.082	0.136	0.145	0.148
12.8	0.182	0.197	0.192	0.126	0.13	0.133	0.082	0.084	0.084	0.084	0.087	0.087	0.166	0.167	0.179
6.4	0.224	0.241	0.241	0.145	0.147	0.15	0.088	0.089	0.089	0.098	0.098	0.099	0.199	0.204	0.215
3.2	0.297	0.31	0.322	0.176	0.181	0.185	0.095	0.097	0.096	0.11	0.109	0.114	0.239	0.278	0.257
1.6	0.413	0.463	0.468	0.242	0.265	0.26	0.118	0.126	0.126	0.131	0.136	0.152	0.313	0.339	0.337
0.8	0.805	0.927	0.899	0.614	0.815	0.714	0.674	0.858	0.883	0.217	0.266	0.287	0.632	0.814	0.928
0.4	1.356	1.551	1.51	1.438	1.623	1.57	1.581	1.712	1.702	1.146	1.372	1.311	1.737	1.857	1.86
0.2	1.794	1.909	1.901	1.814	1.965	1.917	1.856	2.033	1.994	1.711	1.873	1.835	2.012	2.092	2.069
0.1	2.068	2.199	2.197	2.092	2.211	2.181	2.103	2.217	2.177	2.106	2.203	2.18	2.117	2.197	2.207

Fig. 5c: HEK293-NFAT, VEGFA-121

DutaFab concentration (nM)	VP mat			VP mat YHE		
25	3168	2907	2972	2906	3077	2651
12.5	2917	3156	2698	2973	3168	2916
6.25	3272	3361	2873	2927	2929	2879
3.13	3157	2853	3212	2946	2965	3524
1.56	3815	3465	3918	3171	2973	3222
0.781	5081	4349	4813	3085	3045	3637
0.391	9782	8814	8976	3760	3529	3620
0.195	11632	10714	11740	11975	12046	12298
0.0977	11435	11359	12779	10881	12942	12633
0.0488	10335	11854	11540	13097	12638	12238
0.0244	10337	11139	11932	13076	12043	12036

Fig. 5c: HUVEC, VEGFA-165

DutaFab concentration (nM)	VP mat			VP mat YHE		
25	238263	234832	224235	232371	225750	221389
12.5	254734	239965	221582	238244	260900	235297
6.25	226229	231303	241742	222733	250644	233942
3.13	245499	223514	224547	215156	249109	229311
1.56	239961	234069	219547	238102	259883	223179
0.781	247170	245191	230616	236537	259425	242870
0.391	308303	284581	274175	239585	241666	240702
0.195	414119	405130	404578	230730	267414	275140
0.0977	473767	476189	455728	439956	468187	440749
0.0488	471317	480205	481908	456573	488174	502801
0.0244	477900	485540	504983	489131	488593	470512

Figure 6a: day 0 treatment; day 6 FA

10 µg VP mat YHE	10 µg anti-VEGF_A	10 µg anti-DIG
2.3	1.5	2.3
2.3	1.8	3
1	2.5	2.5
2.3	2.7	3
2.3	2.5	2.2
1.3	2.5	2.7
1	2.7	1.8
1.7	2	
1.7	1.3	
1.7	2.3	
1	2.2	
2.5	3	
1.2	1.7	
2	2.8	
	1.5	
	2	

Figure 6b: day -3 treatment; day 6 FA

10 µg VP mat YHE	10 µg anti-VEGF_A	10 µg anti-DIG
2.2	2.2	2.3
1.8	0.7	3
2.3	2	2.5
0.5	1.3	3
1	1	2.2
1	2.8	2.7
0.3	2.2	1.8
1.8	1.7	
2.7	2	
2	1.8	
0.5	2.2	
1.5	2.5	
1.2	1.3	
1	2.3	
	1.3	
	1.7